# Evaluation on the Restoration Effects in the River Restoration Projects Practiced in South Korea

Ji Hong An [1], Bong Soon Lim [2], Jaewon Seol [2], A Reum Kim [3], Chi Hong Lim [4], Jeong Sook Moon [5] and Chang Seok Lee [2,*]

1   Korea National Baekdudaegan Aboretum, Bonghwa 36209, Korea
2   Department of Bio & Environmental Technology, Seoul Women's University, Seoul 01797, Korea
3   Division of Forest Ecology, National Institute of Forest Science, Seoul 02455, Korea
4   National Institute of Ecology, Seocheon 33657, Korea
5   National Institute of Environmental Research, Incheon 22689, Korea
*   Correspondence: leecs@swu.ac.kr; Tel./Fax: +82-2-970-5822

**Abstract:** This study evaluated the effects of the restoration of rivers carried out by the central government on streams located in major cities in South Korea. The effects of the restoration were evaluated based on the morphological and ecological characteristics, species composition and richness of vegetation, and a Riparian Vegetation Index of the restored streams. The naturalness of the streams, based on both the morphological and ecological characteristics, as well as the Riparian Vegetation Index of the restored streams was significantly lower than that of the reference rivers. The vegetation profiles of the restored streams did not reflect the flooding regimen of the river. Furthermore, the herbaceous plants found on the streambanks give way to shrubs and then to tree-dominated vegetation, respectively. The species composition of the vegetation in the restored streams showed a significant difference from that of the reference streams and this difference was particularly more significant with regards to the herbaceous plant-dominated vegetation types. The species richness of the restored streams showed a difference among the different streams but was lower than that of the reference streams. The ratio of exotic and gardening plants occupied in the species composition of the restored streams tended to be higher than that in the reference streams. Considering the above results, the restoration effects were usually low in the restored streams. Accordingly, an active adaptive management plan was recommended to improve those problems.

**Keywords:** evaluation; reference information; restoration effect; river restoration; South Korea





## 1. Introduction

In the past, most river floodplains were transformed into rice fields and high banks were constructed along the waterways in order to prevent flooding in Korea, where people depend on rice as a food source. Consequently, the widths of most rivers were greatly reduced. More recently, many rice fields were transformed into urban areas, and naturally meandering and complex channels were forced to become straight and monotonous. In such continuing transformation processes, the riparian vegetation has degenerated greatly or been destroyed altogether by tree cutting, the introduction of exotic species, the diversion and channeling of waterways for agriculture, the use of river beds and shores for cultivation, or for the construction of roads. The rapid decline of those valuable ecosystems has made riparian restoration a focal issue in the public eye, but progress to control the decline has been marginal, which is partially caused by the lack of experience in repairing damaged riparian ecosystems [1–3].

The ecological restoration has been considered in order to improve the ecological productivity in degraded lands, to conserve the biological diversity, and to mitigate the loss of ecosystems [3–6]. Furthermore, the ecological restoration is an opportunity to test ecological theories in the field [4,7]. In these respects, the ecological restoration should go

beyond a 'simple landscape exercise' and should apply ecological models and theories to the restoration practice [2,4,8–11].

Furthermore, the ecological restoration should take into account environmental changes such as climate change, by preventing disasters caused by climate change or by mitigating the consequences by utilizing the ecological service functions displayed by the restored ecosystem [12–15]. In addition, the rivers with riparian vegetation can contribute to alleviating climate change [2,16]. In this regard, securing the whole spatial range of the river, which includes the stream and riparian ecosystems, is the most urgent and necessary task for the ecological restoration of the river [2,10,15].

Numerous restoration projects are carried out worldwide every year, but the success or failure of these restorations is little known due to the absence of any comprehensive evaluation [17–20]. The evaluation of the restoration effects can provide operational feedback and guidance for future ecological restorations and adaptive management [21,22]. However, if a comprehensive evaluation of a restoration project is not undertaken, science becomes restricted from the lack of information. Projects cannot move forward without the knowledge acquired from previous studies. Small investments in networks that facilitate standardized monitoring, information delivery, and evidence-based evaluation will reward us with great help in planning future projects. For example, the policymaking tools with accurate information, the cost, and the space dependence on the possibility of the project's success, can help determine the priority of restoration activities [23]. This investment in the evaluation will also contribute to improving the relevance and applicability of the ecological research for the restoration [24,25]. In this respect, the evaluation of the restoration effect is an essential task for the development of the ecological restoration.

As many scholars are aware of the importance of the evaluation of the restoration effects, in recent years, a series of indicators and methodologies have been developed in order to provide different monitoring strategies [5,6,20,26,27]. Hobbs and Norton [28] suggested assessing the restoration success as a key process of the ecological restoration, along with the identification of the degradation process and its management, determining realistic goals and successful measurement methods, and integrating them into land management and planning strategies. Since then, debates on the goals of restorations [29,30], the influence of climate change [9,31,32], and the impact of the socioeconomic environmental changes [33–36] have continued, and the results of such efforts have led to the development of indicators in order to evaluate the success of the restoration [37].

The Society for Ecological Restoration (SER) International Primer on Ecological Restoration made a key contribution for successful restorations as it provided a list of nine key attributes as the appropriate indicators for successful restorations [5] and presented the reference ecosystem or conditions of the reference ecosystem as the target for comparison. SERI and PWG [5] presented the native species-centered species composition, secured the functional groups necessary for continuous development and stability, physical stability, maintenance of normal functions, removal of factors threatening the health and integrity of the restored ecosystem, harmonious integration into a larger ecological matrix, resilience to stress events, and self-sustainability as attributes for comparison with the reference ecosystem. In addition, SERI and PWG [5] suggested three strategies for conducting the evaluation: direct comparison, attribute analysis, and path analysis. In direct comparison, the set variables are compared between the reference and restored sites. In attribute analysis, the attributes are evaluated in conjunction with the list presented in the achievement criteria. In this strategy, the quantitative data obtained through systematic monitoring and other surveys are useful in determining the degree to which each goal has been achieved. Path analysis is a strategy used to interpret the data that is to be compared. In this strategy, the data collected periodically from the restored sites are graphically prepared in order to understand the trends. The changing trends toward the reference conditions prove that the restoration is following its intended path. In recent years, McDonald et al. [6] and Gann et al. [26] provided the progress evaluation 'recovery wheel' as a system, which

was systematized and underpinned the list, in order to evaluate the progression along a trajectory of recovery of the restored ecosystem [26,38,39].

However, most river restoration projects have used the taxonomic diversity of invertebrates or fishes as indicators to assess the progress and success of the restoration projects [40–50] without any explicit explanation regarding how the factors relate to determining the success of the restoration projects [44,45,48]. In some cases, the habitat changes [51–62] were measured and the amenity was also evaluated [63–65]. As a desirable method of evaluating the restoration effect, in recent years, the number of studies comparing the integrity, diversity, and sustainability of the restored ecosystems with the ecological conditions of the reference ecosystems have increased [1,2,66–69]. In addition, various methods and indices have been proposed in order to evaluate the riparian conditions of rivers [3].

This study aims to assess the ecological effects of the river restoration aimed at improving the overall system of the urban rivers in major cities in South Korea, which have been under various human interventions for a long time and returning them to a diverse and sustainable ecosystem. Furthermore, another goal of this study is to generate an improvement plan in order to guide the restored streams toward filling the conditions for a successful ecological restoration.

In the present study, we carried out a vegetation survey and compared the results with those from the natural reference streams in order to assess the effects of the restoration. In addition, we evaluated the naturalness of the restored streams based on the morphological and ecological characteristics of the rivers and the riparian vegetation.

## 2. Study Areas

In order to evaluate the effects of the river restoration that the central government (Ministry of Environment and the Ministry of Land, Infrastructure, and Transport) carried out in South Korea, the Hwangguji, Osan, Musim, Daejeon, Jeonju, Gwangju, Sin, Oncheom, and Changwon streams were selected. The Hwangguji, Osan, Musim, Daejeon, Jeonju, Gwangju, Sin, Oncheom, and Changwon streams are located in Suwon, Osan, Cheongju, Daejeon, Jeonju, Gwangju, Daegu, Busan, and Changwon, South Korea's major cities, respectively (Figure 1). Those river names are abbreviated as HG, OS, MS, DJ, JJ, GJ, S, OC, and CW hereafter, respectively.

The reference streams were selected as the streams that belong to the same watershed as the restored streams and have similar ecological characteristics. However, in South Korea, most streams are degraded due to excessive use (Lim et al., 2021, refer to Figure 1). Therefore, the reference streams were selected as they are located in the remote areas where they avoided any excessive artificial interference. They retain a relatively intact riparian vegetation, that appear in the order of grassland, shrubby forest, and tree forest reflecting the flooding regimen far from the waterway, and thus show a higher naturalness compared to the other streams in South Korea (Figure 1). The Sami stream, Yudong stream, Yongsu stream, Miho stream, Geum river, and Seomjin river were selected as the reference streams and rivers for the Osan, Hwangguji, Daejeon, Musim, Jeonju, and Gwangju streams and the Nakdong river was selected as the reference river for the Sin, Changwon, and Oncheon streams (Figure 1).

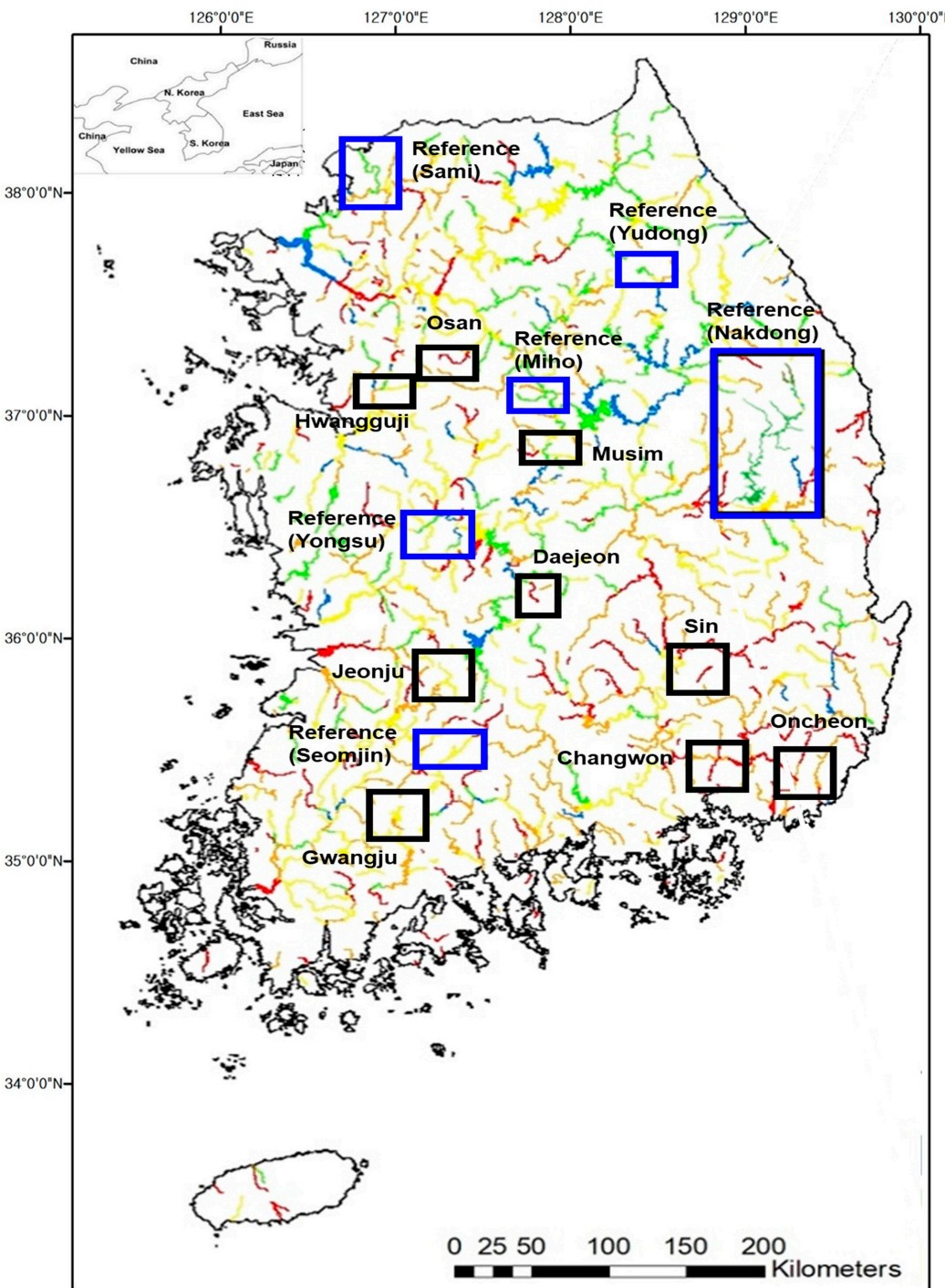

**Figure 1.** A map showing the areas where the streams selected for the evaluation of the restoration effects are located. Dot names indicate the river name. The colors on the map represent the naturalness grade evaluated based on the Riparian Vegetation Index [3].

## 3. Materials and Methods

The restoration effects were evaluated based on the naturalness obtained by synthesizing the morphological (e.g., watercourse sinuosity, diversity of watercourse breadth) and ecological (e.g., land-use on floodplain) characteristics of the river. In addition, the restoration effects were also evaluated based on the riparian vegetation, which is one biological component in the riverine environment and at the same time functions as a habitat environment for other organisms. The evaluation based on the riparian vegetation was

carried out by comparing the species composition and richness and the Riparian Vegetation Index (RVI, [3]) with the reference river.

### 3.1. Evaluation of Naturalness

Naturalness, which means the degree to which it resembles a natural river, was assessed based on the naturalness assessment guidelines developed by synthesizing the morphological and ecological characteristics of a river (Table 1, [2,70]). We classified the traits of a restoration project into the morphological and ecological characteristics and subdivided each trait into five levels: 'very good (5)', 'good (4)', 'medium (3)', 'poor (2)' and 'very poor (1)' [2].

**Table 1.** The degrees of naturalness assigned to the restored and reference rivers. CW: Changwon stream, DJ: Daejeon stream, Gwangju stream, HG: Hwangguji stream, JJ1: Jeonju stream 1, JJ2: Jeonju stream 2, MS: Musim stream, OC: Oncheon stream, OS: Osan stream, SN: Sin stream, Ref: Reference river. 5: very good, 4: good, 3: medium, 2: poor, 1: very poor.

| Item | HG | OS | MS | DJ | JJ1 | JJ2 | GJ | S | OC | CW | Ref |
|------|----|----|----|----|-----|-----|----|----|----|----|-----|
| Sinuosity of watercourse | 1 | 1 | 1 | 1 | 1 | 1 | 1 | 1 | 1 | 1 | 5 |
| The number of sandbars | 1 | 1 | 1 | 1 | 1 | 1 | 1 | 1 | 1 | 1 | 5 |
| Diversity of flow | 1 | 1 | 2 | 2 | 1 | 1 | 1 | 1 | 1 | 1 | 5 |
| River profile | 2 | 2 | 2 | 2 | 2 | 2 | 1 | 1 | 1 | 1 | 4 |
| Diversity of water course breadth | 1 | 1 | 2 | 2 | 1 | 2 | 1 | 1 | 1 | 1 | 4 |
| Degree of waterfront protection | 1 | 1 | 1 | 1 | 1 | 1 | 1 | 1 | 1 | 1 | 4 |
| Artificial degree of bank | 2 | 2 | 1 | 1 | 2 | 2 | 1 | 2 | 1 | 2 | 4 |
| Land use within bank | 2 | 2 | 2 | 1 | 2 | 2 | 1 | 1 | 1 | 2 | 5 |
| Floodplain use | 2 | 2 | 2 | 2 | 2 | 2 | 1 | 2 | 1 | 2 | 4 |
| Transverse artificial facilities | 2 | 2 | 2 | 2 | 1 | 2 | 2 | 2 | 1 | 2 | 4 |

### 3.2. Vegetation Survey

The vegetation map was made based on the interpretation of aerial photographs and field checks. The aerial photo images were used in order to identify the vegetation types and the landscape boundaries. These vegetation types and landscape elements were confirmed by field checks. The landscape attributes were overlapped onto topographical maps at 1:5000 scales. The patches smaller than 1 mm on the map were excluded from this study because of the uncertainty of their sizes and shapes [71]. The mapping was performed using the ArcView GIS (Geographic Information System), and the landscape ecological analyses were conducted with the ArcView GIS software.

A vegetation profile was prepared by carefully depicting the microtopography and major plant species in a belt transect installed in 10 m widths between embankments on both sides of the river. The vegetation samples from the restored streams and reference rivers were compared using several metrics including the species composition, species diversity, and the percentage of exotic species [2].

The vegetation survey was conducted during the summer (i.e., June to August) by recording the cover class of plant species appearing in quadrats of 2 m × 2 m, 5 m × 5 m, and 20 m × 20 m size in grassland, shrub-land, and tree dominated stands, respectively and installed randomly in the riparian zone of the restored streams selected for the study [72]. The dominance was estimated with the Braun–Blanquet [73] ordinal scale from 1 (75%). The nomenclature followed Lee [74] and Korea National Arboretum [75].

### 3.3. Data Processing

Each ordinal cover scale was converted into the median value of the percent cover range in each cover class. The relative coverage was determined by dividing the cover fraction of each species by the summed cover of all species in each plot and then multiplied 100 to the value. The relative coverage was regarded as the importance value of each species [76]. A matrix of importance values for all of the species in all of the plots was

constructed and used as data for the ordination using the detrended correspondence analysis (DCA; [77]).

The naturalness, based on the riparian vegetation, was evaluated from the perspectives of each vegetation component including species diversity, community diversity, vegetation profile, and the ratios of the number of exotic, obligate upland, and annual plant species, and the Riparian Vegetation Index obtained by incorporating the results evaluated on each component [3]. The community diversity was obtained from the vegetation maps and based on the number of plant communities expressed on the vegetation map. The ratios of the number of exotic, annual, and obligate upland plant species were obtained from the percentage of the number of species to the total number of species. The naturalness, based on the vegetation profile, was evaluated based on the response of the vegetation to natural and artificial disturbances, according to Lee and You [78]. If the vegetation profile has an all vegetation zone including herb, shrub, and tree-dominated zones, and they are dominated by native species and the tree-dominated zone is dominated by non-native or upland species, 5 (highest) and 4 scores were provided, respectively. If the vegetation profile is composed of herb and shrub-dominated zones without the tree-dominated zone, 3 scores were provided. If the vegetation profile is composed of just herb-dominated zone, and they are dominated by both perennial and annual plants and by only annual plants, 2 and 1 (lowest) scores were provided, respectively. The species diversity was based on the number of species surveyed. The score for each vegetation component, ranging from 1 (lowest) to 5 (highest) was provided by dividing, at the same interval, the range between the highest and lowest values of each component collected at regular intervals throughout the country [3]. The weighted values of each vegetation component were determined with the aid of experts who participated in a national project in order to evaluate the integrity of the rivers' ecosystems. A weighted value of two points was given to the percentage based on the number of species of exotic, annual, and obligate upland plants, as well as species diversity. We assigned weighted values of four points for community diversity, which addresses the composite factor related to various vegetation types as a two-dimensional component. The vegetation profile expresses the horizontal and vertical diversities of vegetation; eight points were conferred to this component. The Riparian Vegetation Index was obtained from the sum of the scores multiplied by the weighted value of each vegetation component. The Riparian Vegetation Index was divided into five grades of "very good", "good", "moderate", "poor", and "very poor" [3]. The species diversity was compared using rank-abundance curves, which graphically depict the patterns of species diversity and dominance [79,80]. The percentage of exotic species was calculated by dividing the number of exotic species by the total number of species [2,3].

*3.4. Statistical Analysis*

The one-way analysis of variance (ANOVA) was used in order to compare the differences in the percentage of the exotic plants between the restored and reference rivers. The difference in values among sites was tested by Tukey's honestly significant difference (HSD) test. The statistical analysis was carried out using SPSS 19. The detrended correspondence analysis (DCA) is an eigenvector ordination technique based on the correspondence analysis (CA or RA). It is especially suited for the analysis of ecological data sets based on sample units and species [77]. The difference in species composition among restored, unrestored, and natural stands was analyzed using the DCA.

## 4. Results

*4.1. Naturalness Degree Based on the Riverine Structure*

With a naturalness scale ranging from 1 to 5, with 5 being the most natural, the naturalness degrees of the restored rivers, based on the sinuosity of a watercourse, the number of sandbars, diversity of flow, river profile, diversity of watercourse breadth, the naturalness of waterfront protection material, artificial degree of the bank, land use within the bank, floodplain use, transverse artificial facilities, and vegetation type were recorded

as 1 or 2, very poor or poor degree in all items except for the vegetation profile (Table 1). Meanwhile, the naturalness degree of the natural reference river was recorded as 4 or 5 in all items (Table 1).

### 4.2. Vegetation Profile

The most dominant plants that make up the riparian vegetation, such as *Salix koreensis*, *S. gracilistyla*, *S. integra*, *S. koriyanagi*, and *Phragmites japonica*, were introduced in the restored rivers but the spatial arrangement of the plants does not reflect the flooding regimen of rivers (Figure 2). Moreover, the plant species used for landscaping, such as *P. serrulata* var. *spontanea*, *Z. japonica*, *M. sacchariflorus*, *C. drumondii*, *C. tinctoria*, etc., were excessively introduced and exotic species, such as *Artemisia selengensis*, *Erigeron annuus*, *Erigeron canadensis*, *Helianthus tuberosus*, etc., occupied large areas. In addition, stagnant aquatic plants, such as *Zizania latifolia* and *Typha orientalis*, that are not suitable for river ecosystems with running water were also introduced and artificial facilities for recreation such as walkways, bikeways, parking lots, and bare grounds were excessively introduced (Figure 2).

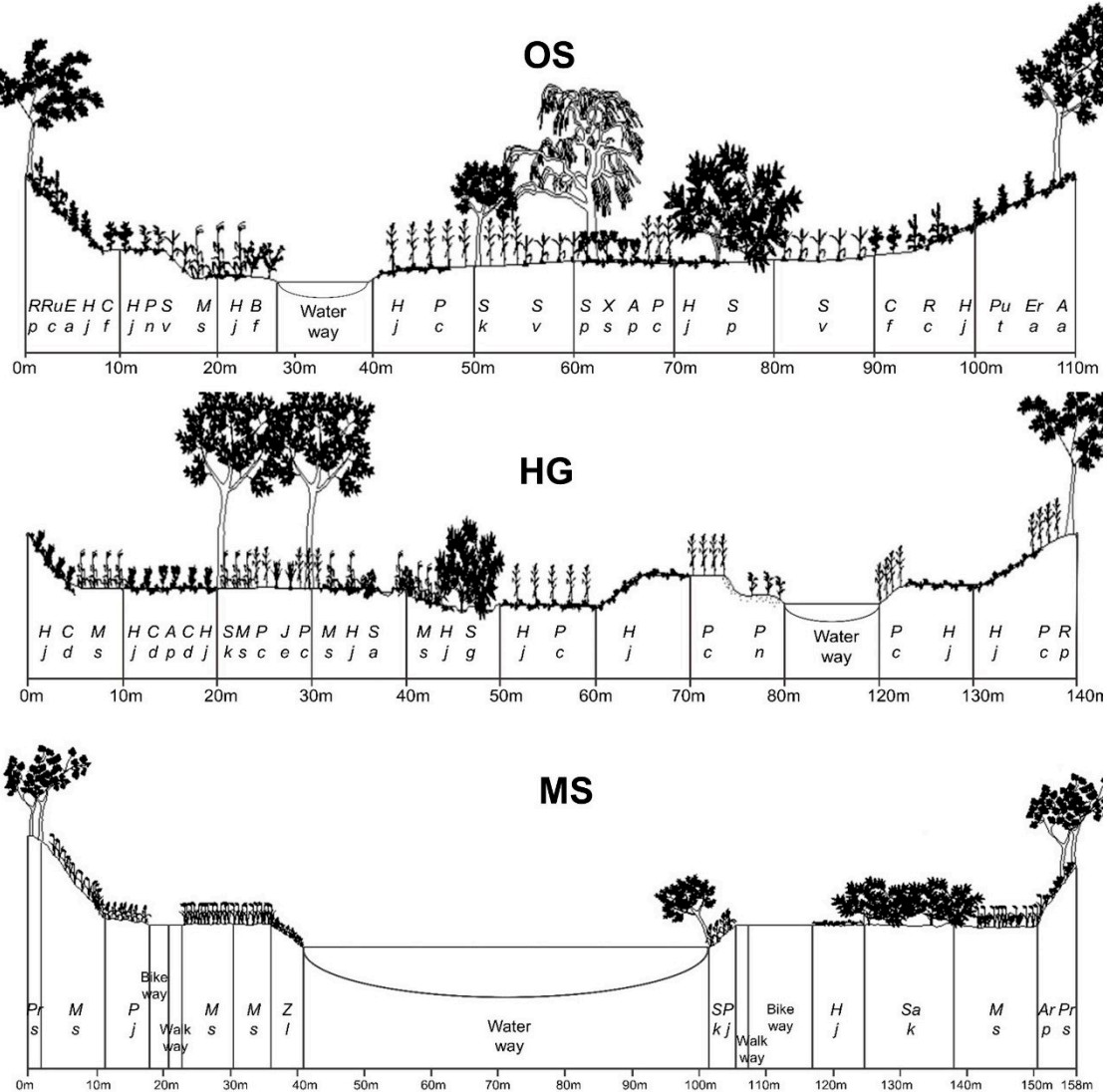

**Figure 2.** *Cont.*

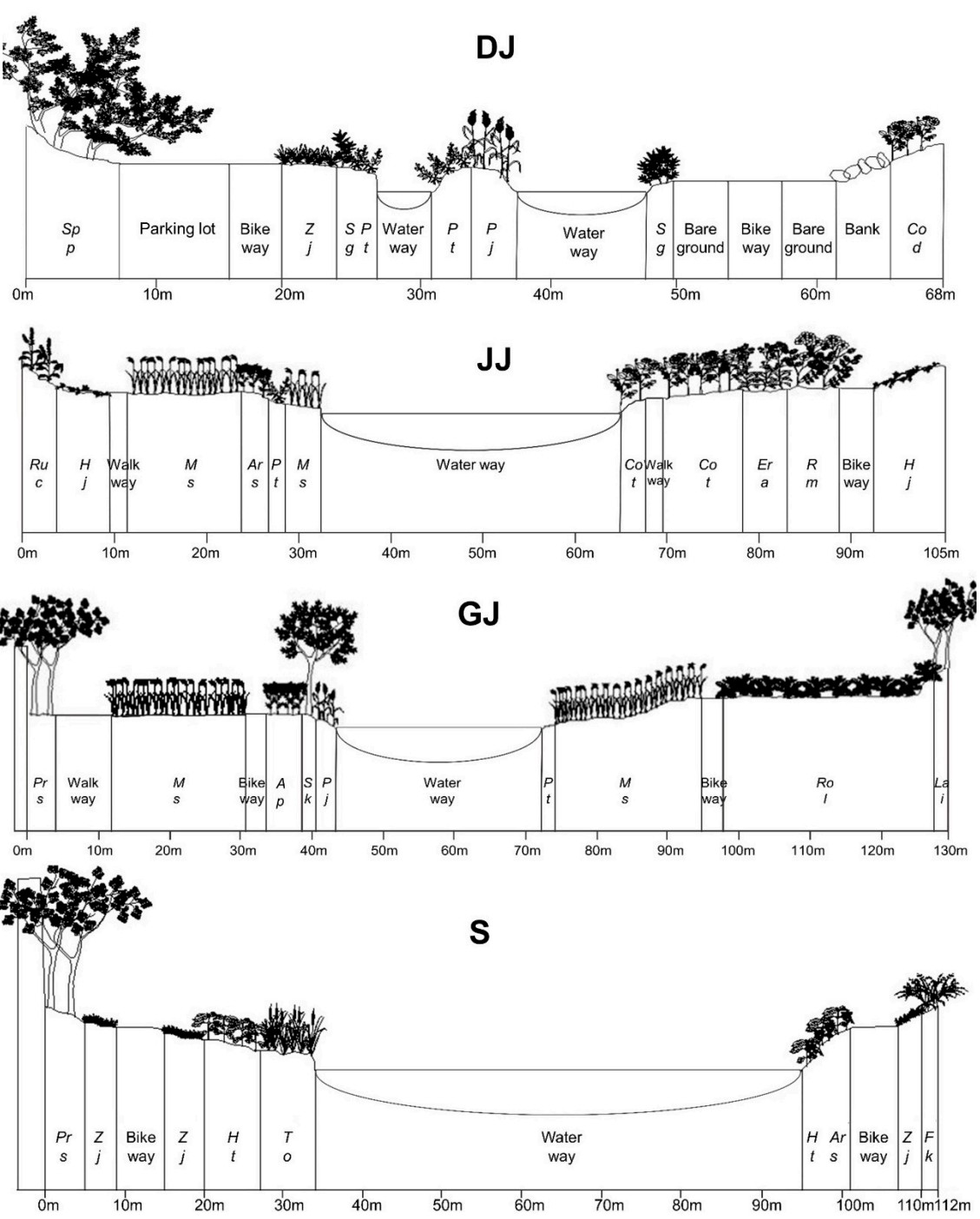

**Figure 2.** *Cont.*

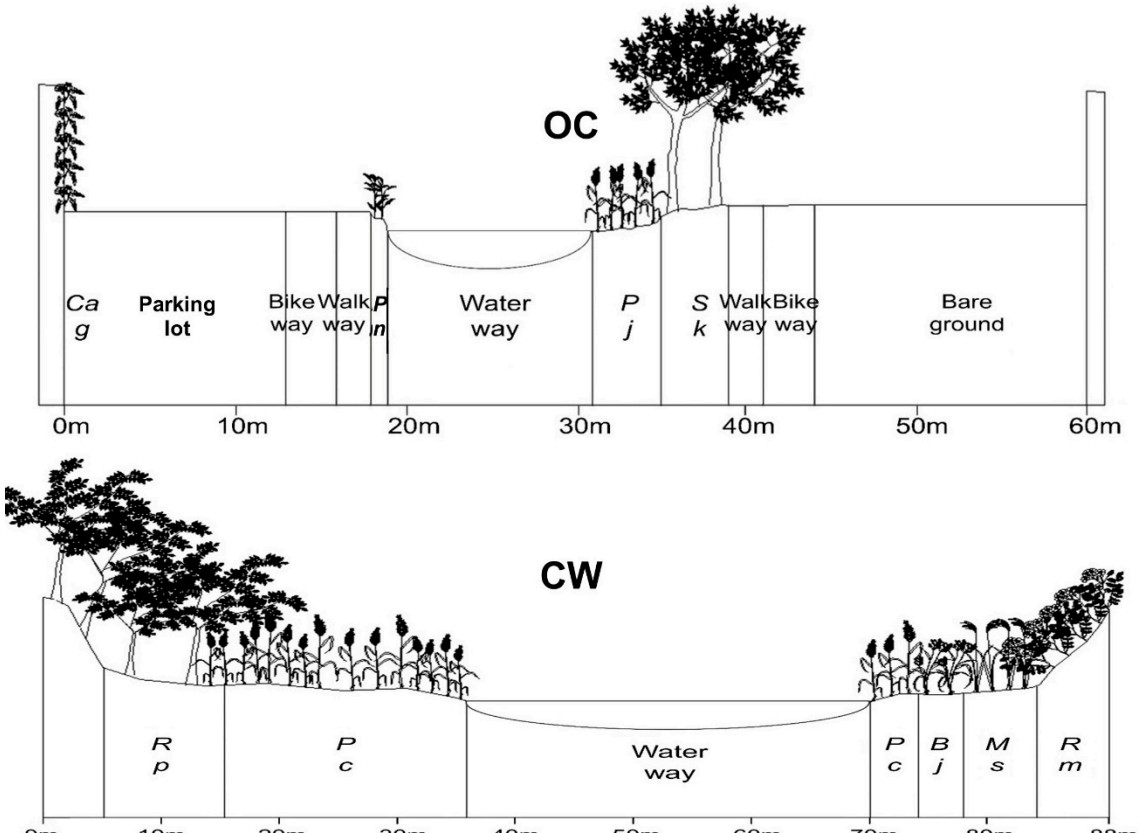

**Figure 2.** Stand profiles of riparian vegetation collected from the restored rivers. CW: Changwon stream, DJ: Daejeon stream, Gwangju stream, HG: Hwangguji stream, JJ: Jeonju stream, MS: Musim stream, OC: Oncheon stream, OS: Osan stream, SN: Sin stream. *Aa*: *Ailanthus altissima*, *Arp*: *Artemisia princeps* var. *orientalis*, *Ars*: *Artemisia selengensis*, *Bf*: *Bidens frondosa*, *Bj*: *Bromus japonicus*, *Cag*: *Campsis grandiflora*, *Cc*: *Commelina communis*, *Cd*: *Carex dimorpholepis*, *Cf*: *Chenopodium ficifolium*, *Cj*: *Calystegia japonica*, *Cod*: *Coreopsis drumondii*, *Cot*: *Coreopsis tinctoria*, *Ea*: *Erigeron annuus*, *Ec*: *Erigeron canadensis*, *Fk*: *Forsythia koreana*, *Hj*: *Humulus japonicus*, *Ht*: *Helianthus tuberosus*, *Je*: *Juncus effusus*, *Lai*: *Lagerstroemia indica*, *Ma*: *Morus alba*, *Ms*: *Miscanthus sacchariflorus*, *Pc*: *Phragmites communis*, *Pj*: *Phragmites japonica*, *Pn*: *Persicaria nodosa*, *Prs*: *Prunus serrulata* var. *spontanea*, *Pt*: *Persicaria thunbergii*, *Put*: *Pueraria thunbergiana*, *Rc*: *Rumex crispus*, *Rm*: *Rosa multiflora*, *Rol*: *Rorippa indica*, *Rp*: *Robinia pseudoacacia*, *Sa*: *Sicyos angulatus*, *Sak*: *Salix koriyanagi*, *Sg*: *Salix gracilistyla*, *Sk*: *Salix koreensis*, *Sp*: *Salix pseudolasiogyne*, *Spp*: *Spiraea prunifolia*, *Sv*: *Setaria viridis*, *To*: *Typha orientalis*, *Xs*: *Xanthium strumarium*, *Zj*: *Zoysia japonica*, *Zl*: *Zizania latifolia*.

Additionally, the riparian vegetation tended to appear in the order of grassland, shrubland, and tree forest far away from the waterway and thus reflected a flooding disturbance regime in the natural reference rivers (Figure 3). A *P. japonica* community and mixed community of *P. japonica* and *Salix gracilistyla* dominated the grassland zone. The shrub zone was dominated by *S. gracilistyla*, *Carex dimorpholepis*, *Acer tataricum* subsp. *ginnala*, etc. and *S. koreensis*, *A. tataricum* subsp. *ginnala*, *Juglans mandshurica*, *Fraxinus rhynchophylla*, etc. dominated the tree forest zone (Figure 3).

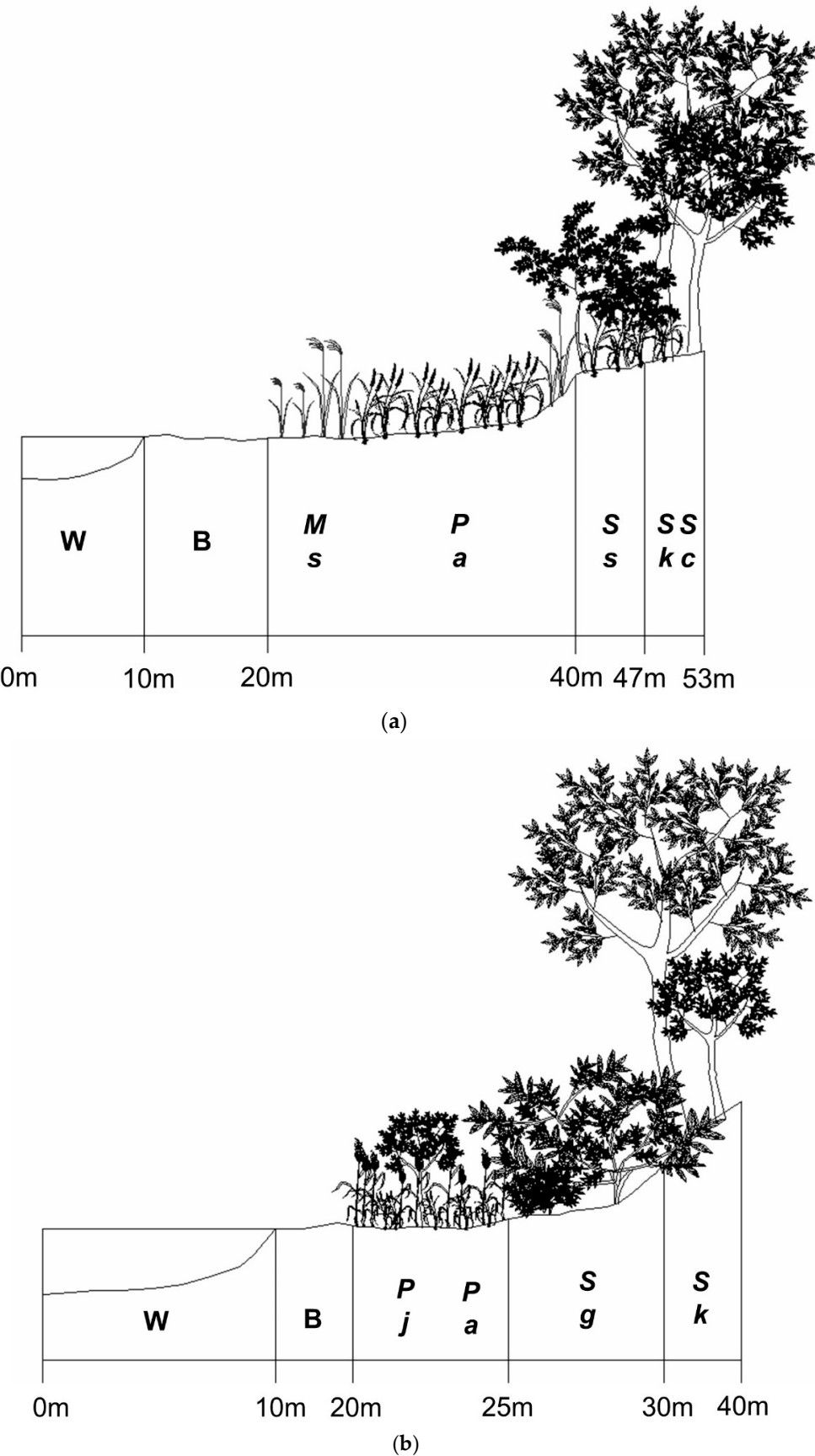

**Figure 3.** *Cont.*

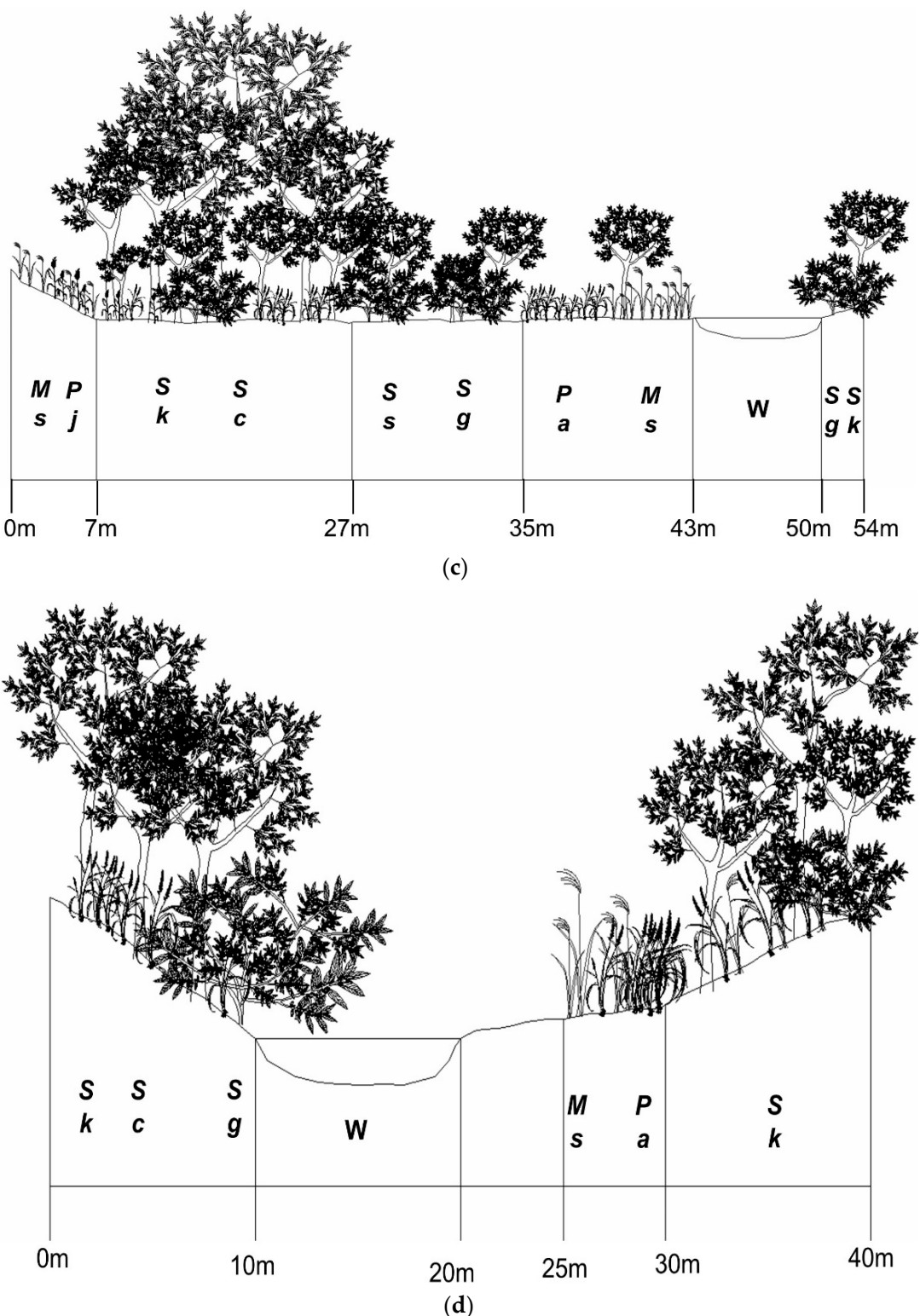

**Figure 3.** *Cont.*

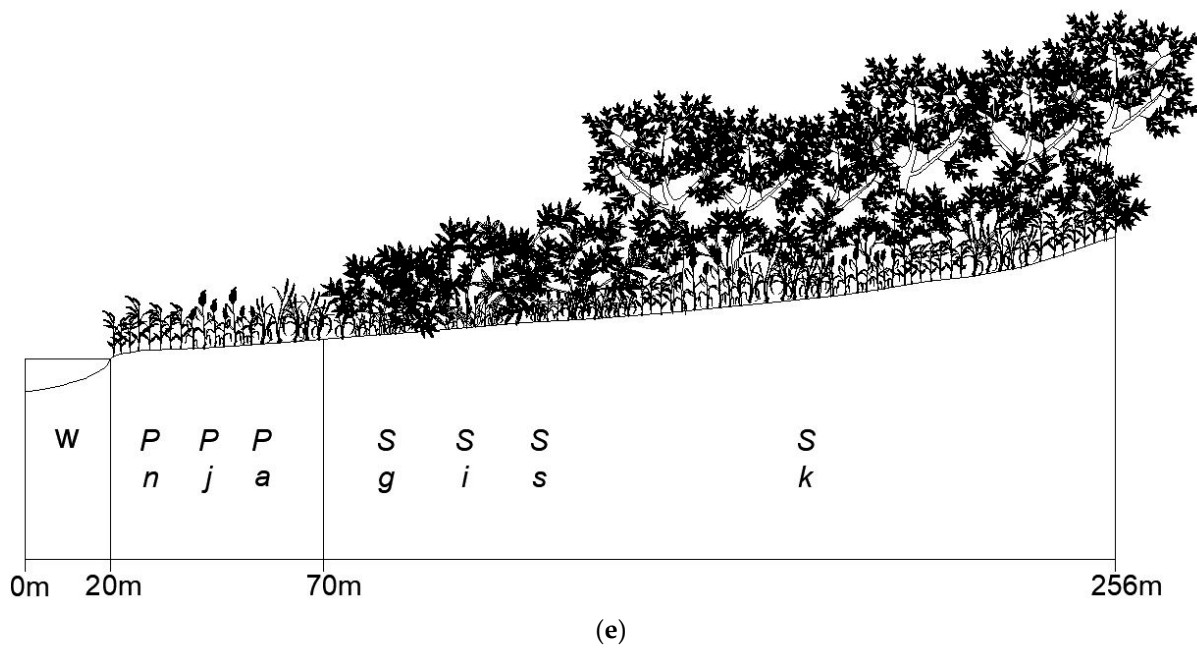

(**e**)

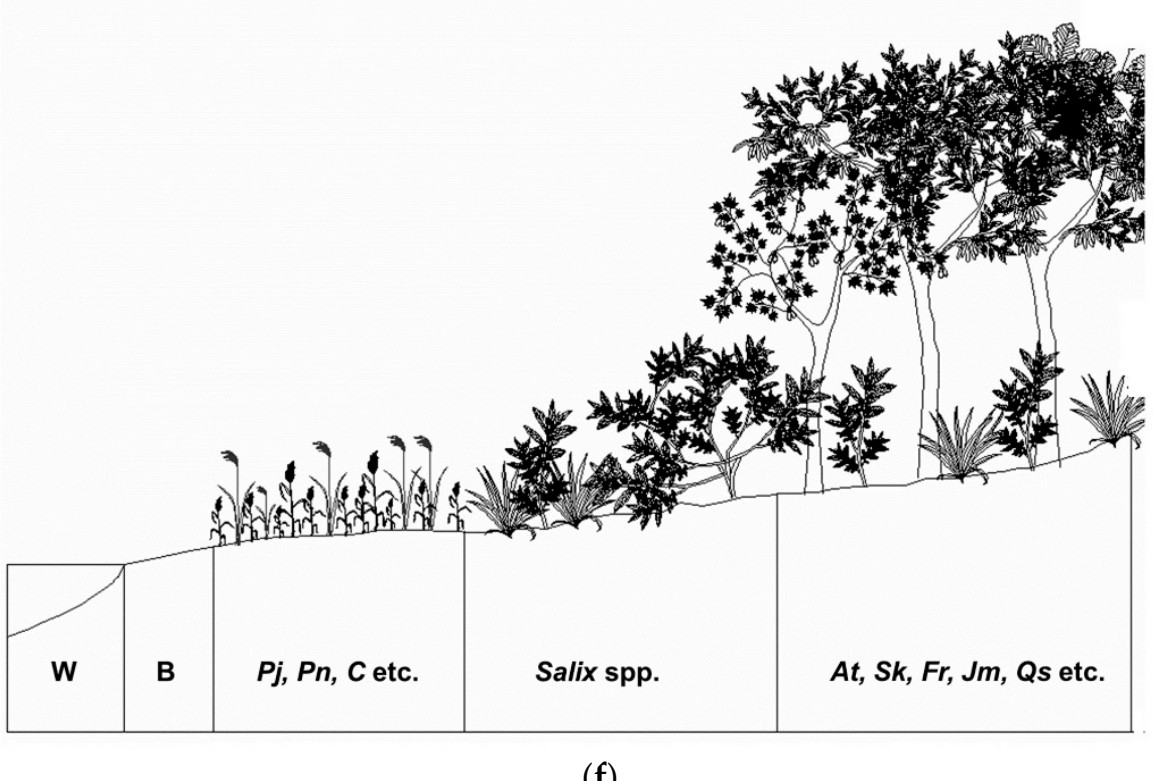

(**f**)

**Figure 3.** *Cont.*

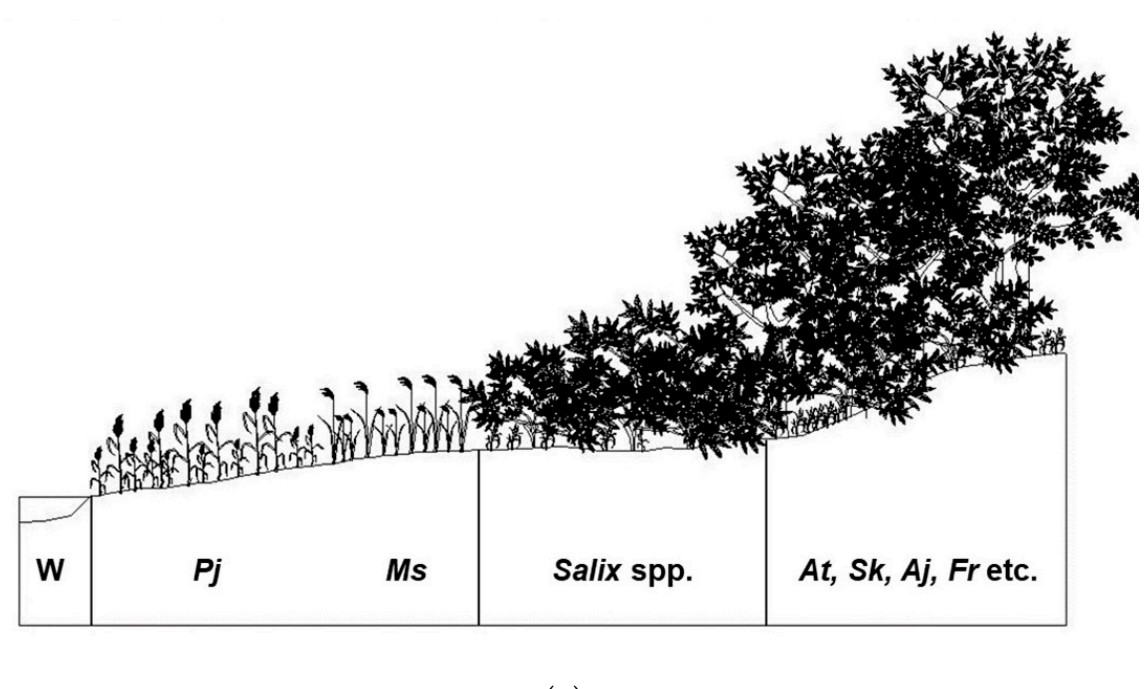

(**g**)

**Figure 3.** The reference information of the restored rivers and streams expressed as the stand profile. The reference information was collected from the Sami (**a**), Yudong (**b**), Miho (**c**), Yongsoo (**d**), Seomjin (**e**), Nakdong (**f**), and the Geum rivers (**g**). The reference rivers and streams were selected as the rivers and streams with a good conservation status as the rivers and streams belonging to the same watershed as the restored rivers and streams (Lim et al., 2021). Aj: *Alnus japonica*, At: *Acer tataricum* subsp. *Ginnala,* B: Bare ground, C: *Carex* spp., Fr: *Fraxinus rhynchophylla*, Jm: *Juglans mandshurica*, Ms: *Miscanthus sacchariflorus*, Pa: *Phalaris arundinacea*, Pj: *Phragmites japonica*, Pn: *Polygonum nodosum*, Qs: *Quercus serrata*, Sc: *Salix chaenomeloides*, Sg: *Salix gracilistyla*, Si: *Salix integra*, Sk: *Salix koreensis*, Ss: *Salix subfragilis*, W: Waterway.

*4.3. Species Composition*

Comparing the species composition of the vegetation between the restored rivers and the reference rivers usually showed a significant difference except for the OC (Figure 4). When comparing the species composition according to the type of vegetation zone, the species composition of the tree dominated vegetation zone that was typically restored with the introduction of *S. koreensis* and the shrub dominated vegetation zone, that was restored with the introduction of *S. gracilistyla*, *S. integra* or *S. koriyanagi* showed a relatively similar species composition. However, the species composition of the herbaceous plant dominated vegetation zone where various plant species used for landscaping were introduced and where many exotic plants had invaded, showed a significant difference (Figure 4).

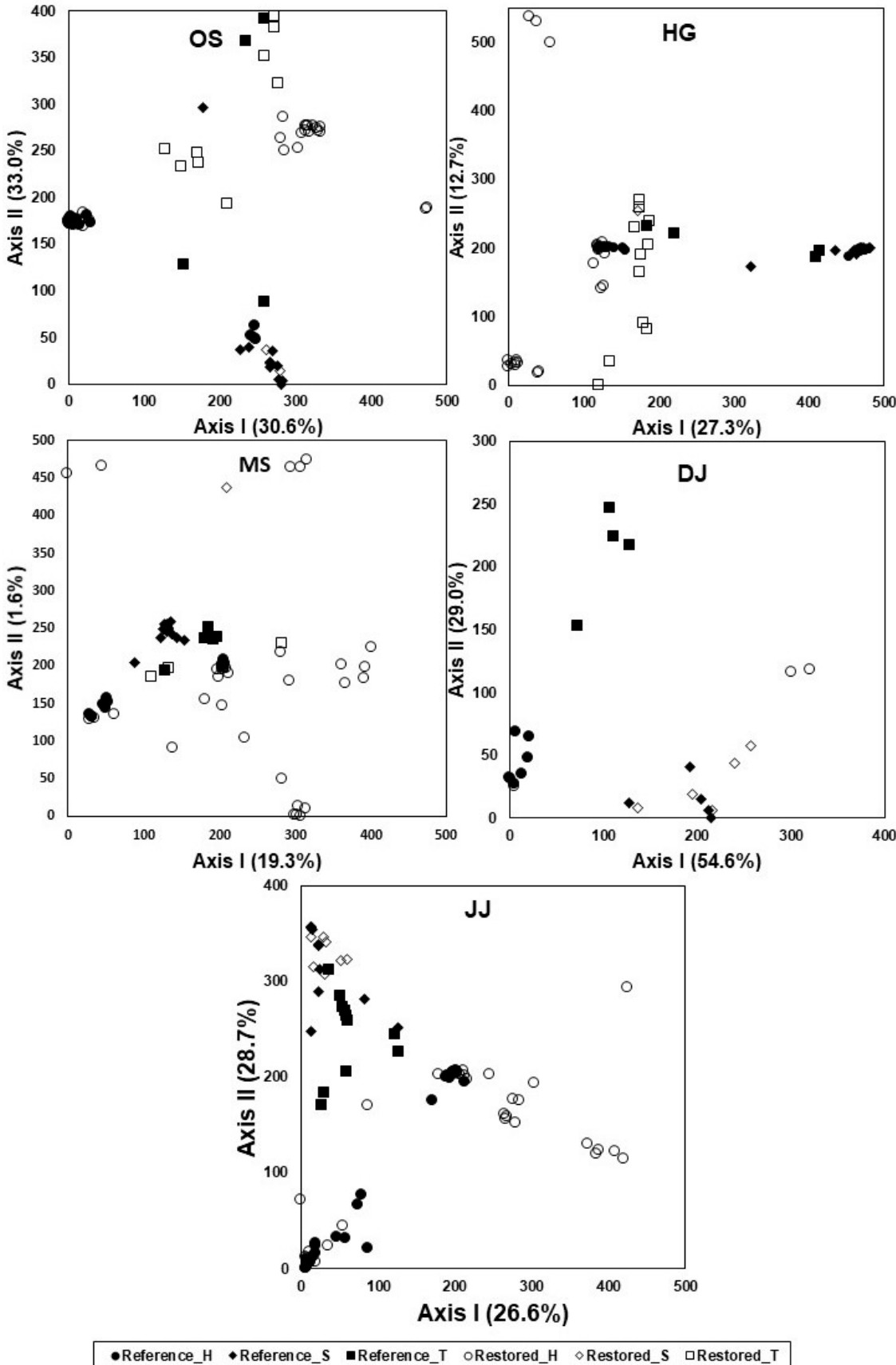

**Figure 4.** *Cont.*

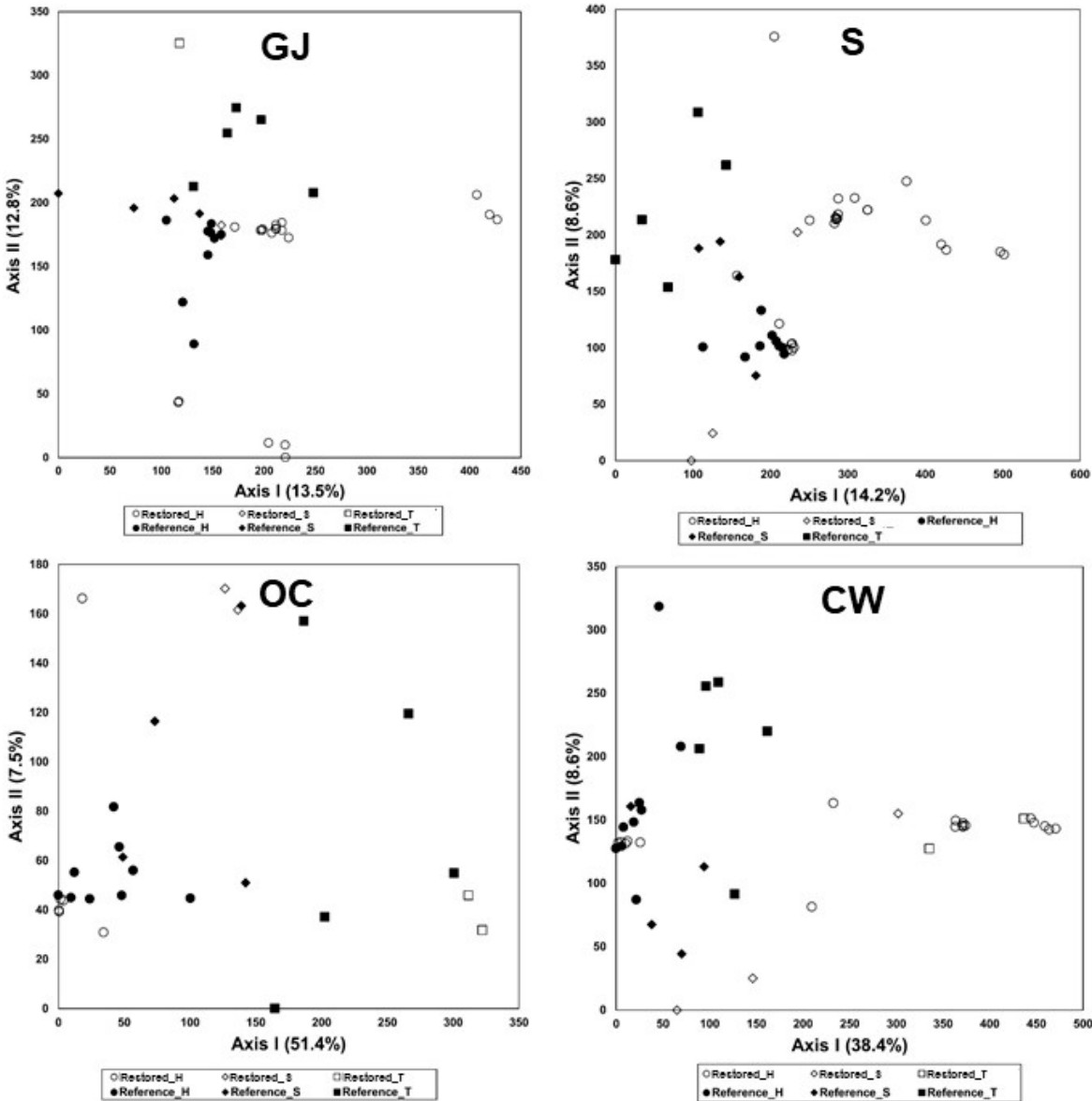

**Figure 4.** DCA ordination of the stands based on the vegetation data collected from the restored and natural reference rivers. Abbreviated river name is the same as in Figure 3. T: tree dominated vegetation zone, S: shrub dominated vegetation zone, H: herbaceous plant dominated zone.

*4.4. Species Richness*

The species richness of the restored rivers was lower than that of the reference river (Figure 5). The species richness showed a large difference among the restored rivers. Comparing the species richness of the restored rivers, the species richness was the highest in the OS and higher in the order of the MS, JJ, CW, S—HG, DJ, OC, and the GJ (Figure 5).

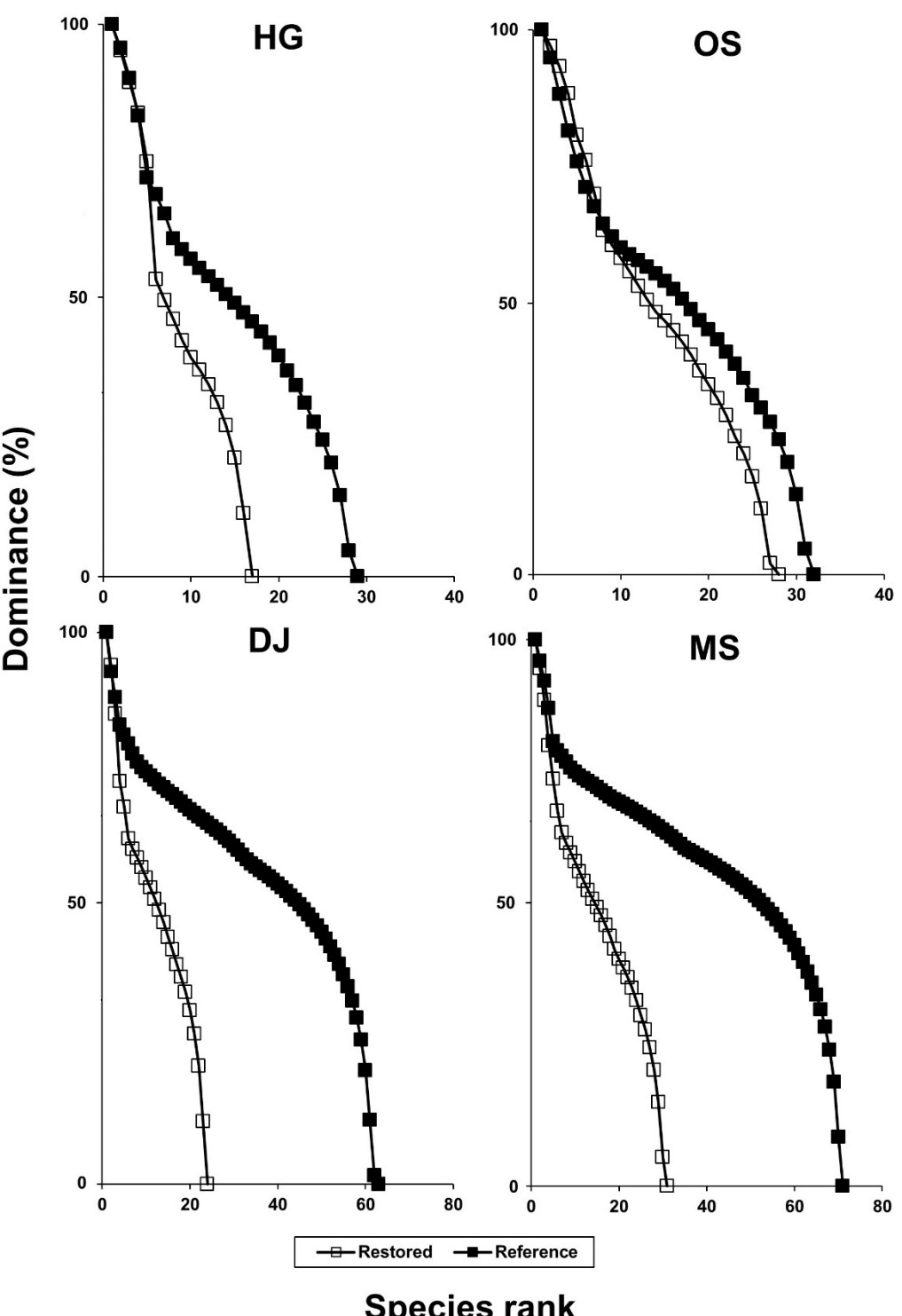

**Figure 5.** *Cont.*

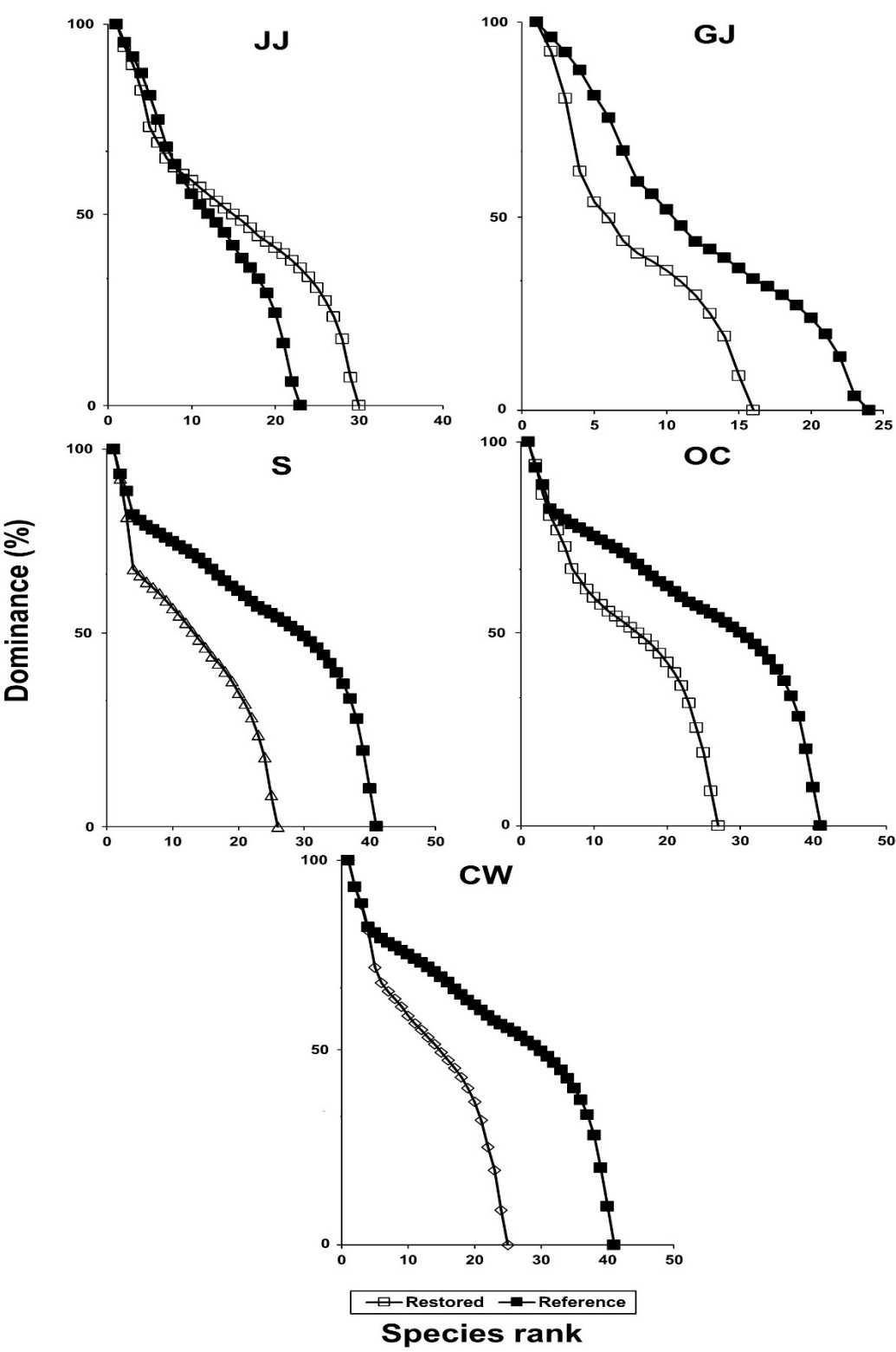

**Figure 5.** A comparison of the species rank-dominance curves of the vegetation among the restored and natural reference rivers. Abbreviated river name is the same as in Figure 3. Reference: natural reference river, Restored: restored river.

*4.5. Exotic Plant Species*

Many exotic plants such as *A. altissima*, *R. pseudoacacia*, *Artemisia selengensis*, *Erigeron annuus*, *Erigeron canadensis*, *Helianthus tuberosus,* etc. invaded the restored streams (Figure 2). Seven streams of the OS, MS, DJ, JJ, GJ, S, and OC, among the nine restored streams had

a higher ratio of exotic and gardening plants than that of the reference river, and the two other rivers were lower (Figure 6). Among the restored rivers with higher foreign species rates, the differences between the MS, DJ, and the JJ and their reference streams were statistically significant.

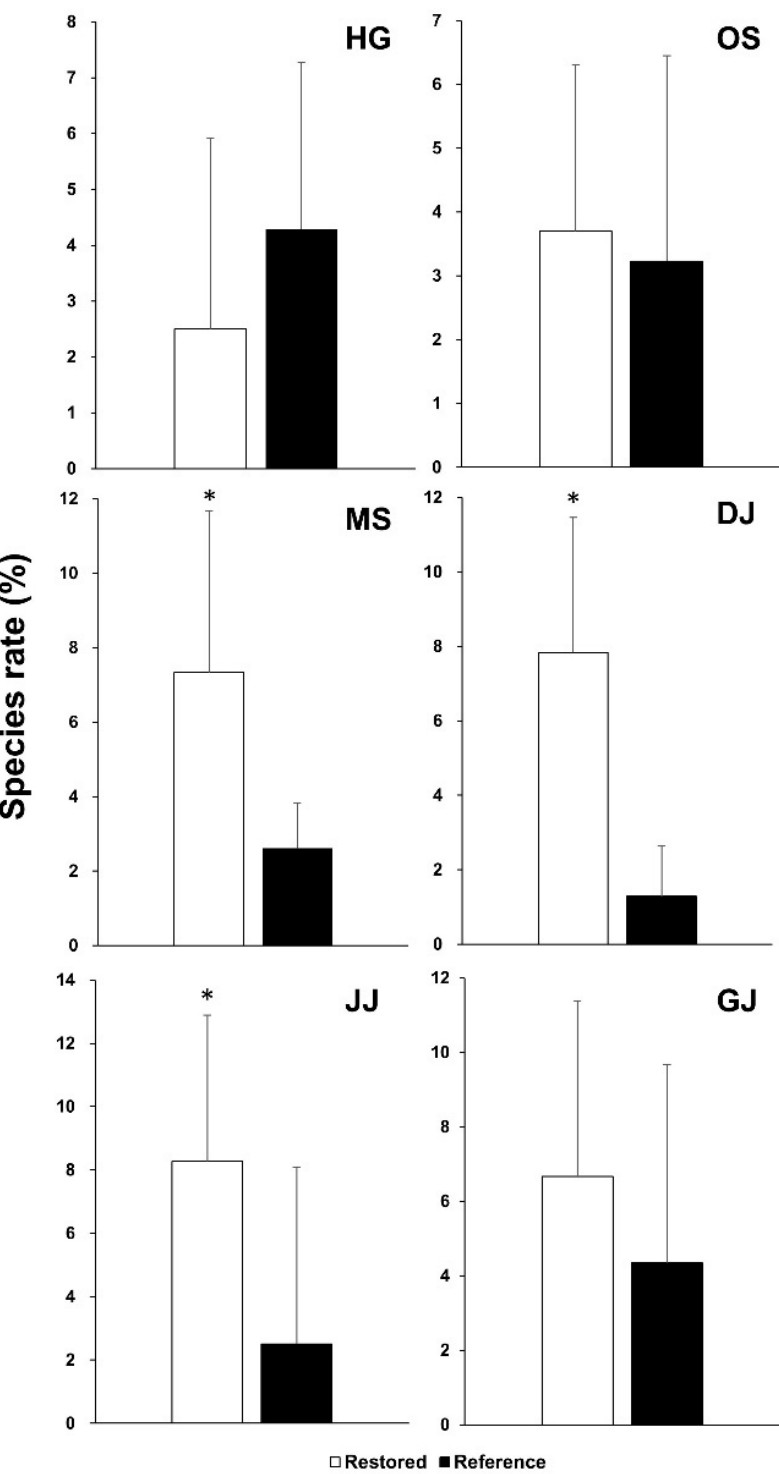

**Figure 6.** *Cont.*

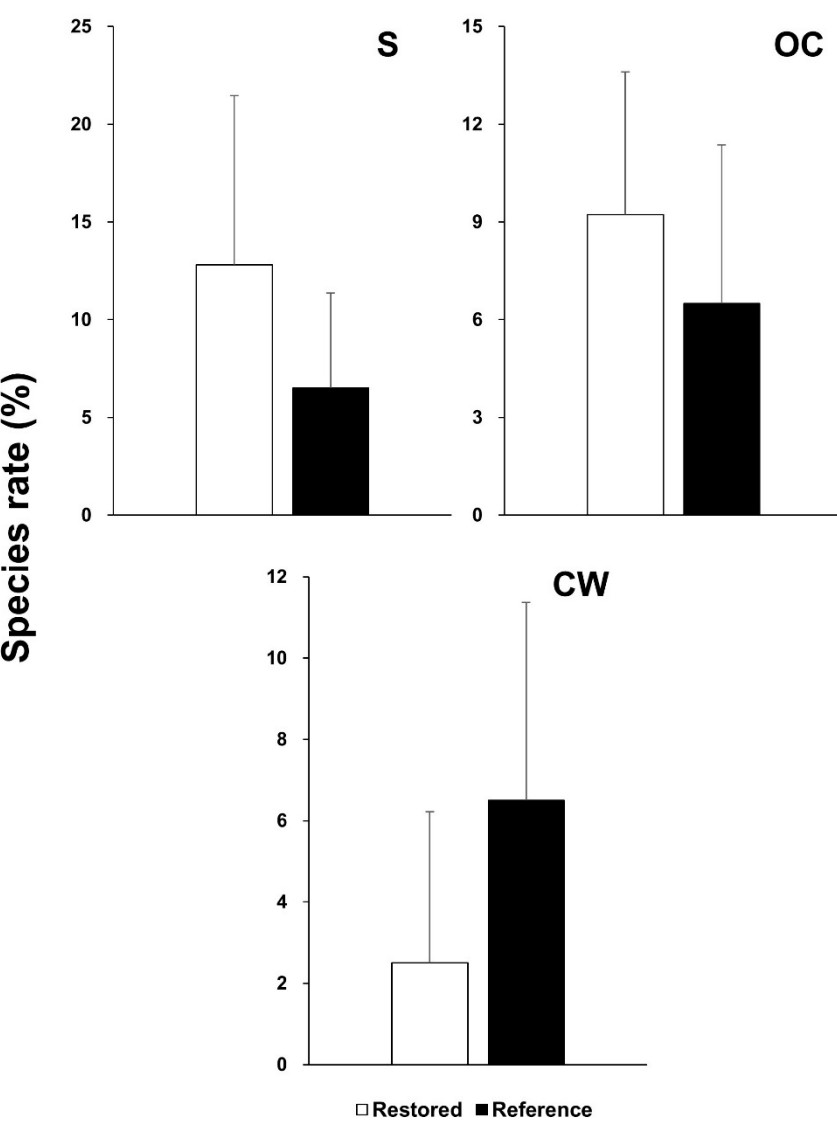

**Figure 6.** Comparisons of the percentage of exotic and gardening plants between the restored stream and the reference stream. The abbreviated stream name is the same as in Figure 2. Asterisk means the significant difference of the statistical test between the restored and the natural reference streams, and no asterisk means no difference statistically. Exotic: exotic plant, Landscaping: plants introduced for landscaping.

*4.6. Naturalness Based on the Riparian Vegetation*

As a result of the evaluation of the restoration effect for nine restored rivers selected throughout the whole national territory of South Korea, the score range of 30 (S, very poor grade) to 70 (OS, good grade) points was found, and the average score was 57.0 points, 'moderate' grade. On the other hand, all of the reference rivers were evaluated as 'very excellent' with more than 80 points (82–92 points).

The S, which received the lowest evaluation, showed high ratios in exotic species and obligate upland plants, a simple vegetation profile, and a low community diversity and species richness. The OS received the highest score among the restored rivers where the evaluation was conducted but had a low community diversity and a relatively simple vegetation profile. Consequently, the restoration effect was not significant.

Overall, the reference information was not used in the process of implementing the restoration, and thus the vegetation profile as well as the spatial distribution of the vegetation did not reflect the flooding regime. That is, they were monotonous and unnatural when compared with the reference rivers. In addition, the restoration plan focused on

the introduction of recreational facilities such as bikeways, walkways, parking lots, and sports facilities which all contributed to lowering the restoration effect. Therefore, it was determined that there was a need to supplement those items identified as contributing to the low score.

## 5. Discussion

### 5.1. Evaluation of the River Restoration Carried out in South Korea

The trajectory of a restoration project may be viewed in terms of the ecosystem structure and function [81,82], both of which are impacted greatly by degradation. The fundamental goal of restoration is to return a particular habitat or ecosystem to a condition close to its pre-degraded state [2–6,8,11,26,83].

To effectively restore degraded areas, or to protect existing high-quality areas, we must be able to define the attributes of "normal", undegraded (or "healthy") habitats as a model [2,3,5,6,11,26,84]. One way of setting a baseline from which to measure the success of the restoration is to define the normal "biological integrity" of a system and then measure the deviations from there. Integrity implies an unimpaired condition or the quality or state of being complete or undivided. Biological integrity is defined as "the ability to support and maintain a balanced, integrated, adaptive biological system having the full range of elements and processes expected in the natural habitat of a region" [85,86].

In order to evaluate a river, the ecological attributes of the river are compared with those from an "undisturbed" reference river [67,69,87,88]. In the present study, we compared the species composition and biodiversity of the restored streams with those of the natural reference rivers, that have a good conservation condition among the rivers belonging to the same water system as the restored rivers.

As a result of evaluating the restoration effects for nine restored rivers selected throughout the whole national territory of South Korea, the cross section of the restored rivers showed a double terraced frame with a steep embankment slope (Figure 2), that is different from that of the natural rivers (Figure 3). The narrow and uniform width of the restored rivers made the micro-topography of the restored rivers simpler and their naturalness diminished. Although the material of the waterfront has been changed from artificial to natural materials, it is difficult to accommodate changes caused by water flow due to the excessive introduction of giant boulders. Therefore, there was a big difference in the spatial arrangement between the vegetation of the restored rivers and the vegetation established by reflecting the flooding regimen in the cross section of the natural rivers. This cross-sectional structure of the restored rivers was reflected in the results of the naturalness degree evaluation based on the riparian vegetation, and thus the naturalness degree based on the morphology of the river was evaluated to be very low (Table 1).

The vegetation profiles of the restored rivers did not reflect the flooding regimen of the rivers as the plant species for landscaping and the artificial facilities for recreation were introduced excessively (Figure 2). The species composition of the vegetation in the restored rivers showed a significant difference from that of the reference rivers, particularly in the herbaceous plant dominated zones (Figure 4). The species richness of the restored rivers was lower than that of the reference rivers although the species richness also showed a large difference among the restored rivers (Figure 5). The ratio of the exotic species and the gardening plants in the species composition of vegetation tended to be higher than that in the reference streams (Figure 6). Consequently, the degree of naturalness of the restored rivers, based on the Riparian Vegetation Index, was also much lower than that of the reference streams (Table 2).

**Table 2.** The result evaluated the restoration effect based on the riparian vegetation for nine restored streams. Abbreviated river name is the same as in Table 1. In the vegetation indices, the numbers in parentheses indicate the weighted values. SR: Species richness, EP: The ratio of exotic plant, OU: The ratio of obligate upland plant, AP: The ratio of annual plants, CD: Community diversity, VP: Vegetation profile, RVI: Riparian Vegetation Index. 5: very good, 4: good, 3: medium, 2: poor, 1: very poor.

| Study Site | SR(2) | EP(2) | OU(2) | AP(2) | CD(4) | VP(8) | RVI |
|---|---|---|---|---|---|---|---|
| HG | 3 | 5 | 4 | 2 | 2 | 3 | 60 |
| OS | 5 | 4 | 5 | 5 | 2 | 3 | 70 |
| MS | 3 | 3 | 4 | 4 | 4 | 3 | 68 |
| DJ | 3 | 2 | 3 | 3 | 2 | 3 | 54 |
| JJ | 1 | 5 | 2 | 3 | 4 | 3 | 62 |
| GJ | 2 | 4 | 4 | 1 | 3 | 3 | 58 |
| S | 2 | 1 | 2 | 3 | 2 | 1 | 30 |
| OC | 2 | 3 | 3 | 4 | 2 | 3 | 56 |
| CW | 2 | 4 | 2 | 5 | 3 | 2 | 54 |
| Ref (SM) | 5 | 3 | 4 | 3 | 5 | 5 | 90 |
| Ref (YD) | 5 | 4 | 5 | 2 | 4 | 5 | 88 |
| Ref (YS) | 5 | 4 | 2 | 5 | 4 | 5 | 86 |
| Ref (MH) | 5 | 3 | 3 | 5 | 4 | 5 | 88 |
| Ref (G) | 5 | 3 | 5 | 3 | 4 | 5 | 82 |
| Ref (SJ) | 4 | 3 | 5 | 5 | 4 | 5 | 87 |
| Ref (ND) | 5 | 5 | 4 | 4 | 4 | 5 | 92 |

The river restoration is an attempt to return a certain river ecosystem to a level that can be maintained and developed on its own within the range of fluctuations of the development process of the ecosystem before any disturbance or in the past [2,3,89]. In order to realize this restoration, the longitudinal direction and cross section of the river should be restored to its original shape as much as possible with the use of an old map. If this is impossible, a wide spatial range needs to be secured in order to induce meandering through the process of erosion and deposition that running water naturally creates [10,70]. However, most of the river restoration projects are carried out on a small scale, and thereby have not secured enough longitudinal and transverse space in order to induce meandering. Furthermore, as the meandering waterway is not formed, the diversity of micro-topography such as riffle and pool is not achieved (Table 1).

Another factor that lowers the naturalness is that most river restoration projects focus on recreational use rather than recovery of nature, thereby resulting in the excessive introduction of artificial facilities such as walkways, bikeways, bare ground, parking lots, etc. (Table 1).

The vegetation should be introduced as a local native species and exotic and non-local species should be excluded. The arrangement of the vegetation should be determined based on the reference information, which reflects the disturbance and flooding regimen of the site [2,3,5,6,10,26,70]. However, the vegetation introduced in most river restorations in Korea so far does not follow this principle [11,14,90]. Therefore, exotic species are introduced, plants appearing in static water ecosystems such as ponds or lakes are introduced into rivers where running water exists, and trees are introduced on waterfront where bare ground or annual plants are established due to frequent flooding disturbance (Figure 2, Table 1). Consequently, the restoration effect is not significant despite the huge cost and energy invested and after the restoration, it may be exposed to a greater flooding risk (Tables 1 and 2, [11,14,90]).

To avoid this error, the floodplain should maintain a gentle slope created by flowing water. Furthermore, the ecological diversity and functions need to be improved by allowing various micro-topographies induced by the water flow during flooding. In addition, it is important to leave a large part of the floodplain to the natural process in order to induce the natural settlement of the vegetation (usually annual or perennial herbaceous plants) without any artificial introduction of vegetation. The introduction of vegetation needs to start in the ground adjacent to the embankment, in order to save cost and energy, and to secure the naturalness along with stability [1,10,70].

At the end of the floodplain where the power of the rushing water weakens, backwater wetlands are usually formed, Cyperaceae plants grow, and shrubby willows such as *S. gracilistyla* and *S. integra* are also often established. Therefore, shrubby willows could be recommended as plants to be introduced into these sites [1,10,70].

Most of the embankments existing in Korean rivers are artificial and must be retreated significantly in the long run [10,70], as part of the river project [91]. Therefore, it is not easy to determine the plants to be introduced there. However, considering the disturbance regimen of the site, it would be the softwood zone (dominated by willow, popular, alder, etc.) or hardwood zone (dominated by elm, ash, some of maple, oak, walnut, hackberry, etc.) [10,70]. On the embankments that have not been dominated by artificial interference for a long time, those plants actually appear frequently [92–96]. Therefore, it is necessary to increase their stability along with their naturalness by introducing plants that makeup the softwood zone or hardwood zone rather than introducing exotic species or gardening plants there as was previously the practice.

The protection of the waterfront should depend on the plants that will be established there in the future. If it is not possible to expect protection by vegetation in the early stages of restoration, the construction methods using traditional natural materials, such as stacking branches of riparian plants, can be used for temporary protection. In the case of cobble stacking, it can be applied because the vegetation is easy to establish between the cobbles, but it is premised that the introduced stones are brought from within the basin and used for temporary protection [1,10,70].

In reviewing these results, it is judged that the low restoration effect is due to the subjective promotion of the restoration in the process of implementing the restoration without using the reference ecological information. Indeed, such an assessment has continued [1,3,14,90,97]. In addition, little assessment of the restoration effect has been made. Therefore, no progress is being made even as the project continues [1,90,96,97].

*5.2. Recommendation for Improvement of Current River Restoration Project*

The Ecological restoration is an ecological technology that seeks to provide habitat for various organisms and to secure the future environment of mankind by repairing the environment that humans have damaged by imitating the system and function of the intact environment [5,6,26,83]. The Ecological restoration means copying nature by studying a system of intact nature. We have to grasp the feature of the intact nature in order to heal the disturbed nature. The reference information collected from places with a complete natural appearance contains such features. Therefore, the reference information becomes the goal of the restoration in the process of establishing a restoration plan, and after the restoration is completed, it becomes a tool in order to assess its success or failure [3,6,14,26,90,98].

The riverine landscape is comprised of the stream and riparian ecosystems. When water flowing in a stream ecosystem goes over channel bank, the riparian ecosystems are formed [99,100]. The riparian ecosystem is the ecotone between the aquatic and terrestrial ecosystems. The riparian ecosystem consists of several fluvial surfaces, including the Channel Islands and bars, channel banks, floodplains, and lower terraces [100]. The riparian ecosystem is divided depending on the flooding regimen [99,100]. The one zone, which is frequently inundated, is subjected to current day fluvial geomorphic processes, and is at an elevation that allows shallow-rooted plants to extract water from the water table. The other zone, which is far from the waterfront and thus inundated less frequently. This zone was formed by past fluvial geomorphic processes, is higher in elevation, and where the vegetation is dominated by deeply-rooted plants capable of extracting water from the underlying alluvial aquifer [99,100].

However, in most Korean rivers, the spatial range of those riparian ecosystems was narrowed greatly due to excessive land uses, including the development of rice fields and the urbanization in the riparian ecosystem. Therefore, it is very difficult to find a river with a complete structure (Figure 3, [3,11]). However, the so-called remote streams, which are far from the city and less influenced by humans, such as the streams selected as the reference

rivers in the study, have a relatively intact system of streams as shown in Figure 3. In upcoming river restoration projects, the information collected through a systematic study of the rivers with a near-natural appearance should be organized and carried out and then used as reference ecological information.

## 6. Conclusions

As a result of the evaluation of the restoration effect, the restored streams in Korea were evaluated with a low naturalness in both terms of the morphological and ecological characteristics of the rivers and the composition and spatial distribution of the riparian vegetation. The diverse plant species were introduced for the vegetation restoration, but the flooding regimen, which is significant in the spatial distribution of the riparian vegetation, was not correctly reflected. Exotic or gardening plant species that were not ecologically suitable for the location, were introduced and thus a measure to improve those problems is required. As the principle of the ecological restoration that imitates nature was not reflected in the restoration plan, the rivers were constructed as a steep slope structure. The waterfront was not designed to accommodate the changes from flooding disturbance, making the micro-topography of the restored streams simpler and the naturalness lower. Overall, these projects could be evaluated as artificial park construction projects that include waterways rather than river restoration projects. In this respect, an active adaptive management plan seems to be needed in order to improve those problems. As an improvement measure, first of all, it is important to grasp the features of the integrated rivers in order to heal the disturbed rivers. That is, we have to prepare the reference information systematized by studying a system of the intact rivers in order to realize the ecological restoration. Second, the river zones should be extended and linked to the surrounding terrestrial ecosystems in order to create an ecological network based on the ecological information from the reference area. Third, more diverse microhabitats should be created within the waterways in order to support a greater biodiversity. The flooding during the rainy season produces various microhabitats and human assistance can aid this natural occurrence. Finally, the species composition and its spatial arrangement were focused on landscaping and recreational use during the restoration; this approach should be changed in order to reflect the principles of the ecological restoration.

**Author Contributions:** Conceptualization, C.S.L.; methodology, J.H.A.; validation, J.H.A. and A.R.K.; formal analysis, J.H.A. and A.R.K.; investigation, C.S.L., J.H.A., A.R.K., J.S., B.S.L., C.H.L. and J.S.M.; data curation, J.S., J.S.M. and B.S.L.; writing—original draft preparation, J.H.A.; writing—review and editing, C.S.L. and J.H.A.; visualization, C.H.L. All authors have read and agreed to the published version of the manuscript.

**Funding:** This work was supported by Korea Environment Industry & Technology Institute (KEITI) through Wetland Ecosystem Value Evaluation and Carbon Absorption Value Promotion Technology Development Project, funded by Korea Ministry of Environment (2022003630002).

**Conflicts of Interest:** The authors declare no conflict of interest.

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
