# Peer review of "Evaluation on the Restoration Effects in the River Restoration Projects Practiced in South Korea"

_water, doi:10.3390/w14172739_

Round 1

Reviewer 1 Report

Thank authors for their interesting paper. The language is fluent, the writing logic is reasonable, and the content is rich. After answering the following questions, the manuscript could be accepted.  In the first paragraph of the introduction section, it is mentioned that agricultural activities and urbanization are the driving forces of river landscape degradation, that is to say, farmers and citizens are the direct stakeholders of river restoration. The protection of their rights should be discussed during the formulation of restoration plans. However, I didn't find this content involved in the discussion. The authors just put forward the concept of restoration based on nature in the discussion. This is very good, but there is no social solution at all. After all, restoration ecology is not just a natural science, but also a social science.

Author Response

Reply to Reviewer 1

Thank authors for their interesting paper. The language is fluent, the writing logic is reasonable, and the content is rich. After answering the following questions, the manuscript could be accepted.  In the first paragraph of the introduction section, it is mentioned that agricultural activities and urbanization are the driving forces of river landscape degradation, that is to say, farmers and citizens are the direct stakeholders of river restoration. The protection of their rights should be discussed during the formulation of restoration plans. However, I didn't find this content involved in the discussion. The authors just put forward the concept of restoration based on nature in the discussion. This is very good, but there is no social solution at all. After all, restoration ecology is not just a natural science, but also a social science.

☞ Rice, the staple food of Koreans, is an aquatic plant. In order to cultivate such a rice, the width of the river has been greatly reduced in Korea. Therefore, most rivers have a narrow width, so the flow rate is fast. Consequently, they cannot secure various micro-topography, so the biodiversity is low. In addition, most of the riparian vegetation was removed because it could damage the growth of rice by shading. Nevertheless, as suggested in this manuscript, the current river restoration introduced too many recreational facilities, and the introduced plants, especially herbaceous plants, focused on trimming their aesthetics. In this reality, since the current river has only a minimum space, the rights of farmers and citizens could not be considered.

Reviewer 2 Report

1. What is the Innovativeness in the study?
2. Which model you had study for evaluation?
3. Which control measures you adopted for your study?
4. Have you developed Risk Model? If yes - how and if now - why
5. Limitations in the study?
6. Length of data used in the study?
7. Any other method for restoration ?
8. Have you considered governing equations?
9. Which parameters you have considered for morphological analysis ?
10. Which technique is used for carry out analysis ?
11. Have u used AHP techniques? If no compare your result with AHP
12. Compare your work with other researchers

Major revision

Author Response

Reply to Reviewer 2

  1. What is the Innovativeness in the study?

1) Evaluating the effects of river restoration at the national level

2) Approaching the restoration effect assessment from the system level beyond the

species level
2. Which model you had study for evaluation?

☞ The model mentioned here means reference information.

  1. Which control measures you adopted for your study?

☞ Korean rivers have long been renovated to cultivate rice, their staple food. In

particular, most of the floodplain was transformed into rice paddies, and the width

was greatly reduced. Therefore, in Korea, it is often forgotten that rivers are a system

that stream ecosystem and riparian ecosystem are combined. Therefore, it is viewed

as control measures to inform its true appearance and reflect it in the

restoration project.

  1. Have you developed Risk Model? If yes - how and if now – why

☞ No, I didn’t. However, previous studies have confirmed the safety of the

restoration model proposed in this study. In particular, since willow trees with

flexibility are the main plants in the vegetation introduced, the impact on flooding is

not expected to be significant.

  1. Limitations in the study?

☞ This study did not include various taxa that make up biodiversity, and did not

contain the effect of improving functions.

However, the vegetation used as an evaluation tool in this study is a producer of

the ecosystem, and moreover, although vegetation is a taxon also plays a role as

an environment for other living things. Therefore, it is considered a suitable

material for evaluating the restoration effect. In this regard, the significance of this

study can be confirmed.

  1. Length of data used in the study?

☞ I can’t understand what your question means. Please explain it some more

 substantially.
7. Any other method for restoration?

☞ In countries such as Korea, where ecological restoration has not yet been fully

established, ecological restoration tends to be considered as scenery trimming

rather than healing damaged nature by imitating an intact natural system.

Of course, there are many types of restoration different in levels and methods as

is shown in Figure 1, but I think the restored system should be a sustainable and

self-maintaining system.

Figure 1. Degraded ecosystems lost their structure and functions. Goals of a restoration project have to be decided as to whether to restore, rehabilitate, or replace the degraded site, or whether the best course of action is no action. The restoration process is an attempt to direct the system back to the original state. Complete restoration would involve return to that state; partial return, or other trajectories would result in rehabilitation or replacement by a different system.

  1. Have you considered governing equations?

☞ No, I haven’t considered it.
9. Which parameters you have considered for morphological analysis?

☞ They are presented in Table 1.

  1. Which technique is used for carry out analysis?

☞ I evaluated the restoration effect based on the primer proposed by SERI (2004),

McDonald's et al. (2016), and Gann et al. (2019).

  1. Have u used AHP techniques? If no compare your result with AHP

☞ This study evaluated the restoration effect by selecting variables that have been

applied in various studies attempted to evaluate the restoration effect. Therefore, I

did not feel the need to apply the AHP technique additionally.

  1. Compare your work with other researchers

☞ As suggested in the discussion of our manuscript, this study analyzed and

compared the results of other researchers. And I think a successful restoration is,

above all, to create a system similar to intact nature. Therefore, in this study, the

effect was evaluated by comparing the morphology and ecology of the restored river

with the morphology and ecology of the river close to nature.

Reviewer 3 Report

Sometimes, I have some difficulties with your language. There are things about which I would need some clarification. Are the whole river stretches rehabilitated, or is it just the urban portions that have been worked on? You state that you observed transversals along the river(s) between the weirs. What weirs? What were the time spans between ends of projects and your evaluation? There were also two references that made me curious – No 70 and 73 respectively. One is about aquatic ecology, and the other about plant sociology. These are things that you barely, or not at all, deal with in your paper. The list of references also shows many dated items. River restoration was not such a hot topic in the previous century, as it has become today.

Why did the government undertake these projects? I doubt that it intended to bring the rivers back to some God given pristine state. However, your evaluation is based on the differences between such a state, and the actual vegetation that you observe and measure. Such a state is, however, not attainable unless you eliminate a great majority of the population, and dynamite the cities. I think that this is, what’s is known as eco-fascism. I am sure that you had more commendable motives. Thus, I think that you need to explain, in some more detail, what or whom that inspired you to undertake this study.

Speaking of plant sociology, I find it a bit problematic that you do not consider this in your study. There are things that promote, or counteract development of healthy ecosystems. The vegetation patterns that you, more or less prescribe, have developed over a long time, and in locations, whose characteristics may be very different from the ones in the study areas. Young ecosystems are much less resilient. That’s one reason that you found so many garden flowers there. This indicates another omission, that I find problematic. You are so obsessed by the look of the ecosystems, that you forget about their functioning. You mention their flood attenuation capacity, but I think that it’s essentially the floodplain that provides this service.

Thus, I suggest that you get some help to make the language a bit more clear. There is a need for a more comprehensive account on why you undertook this study, and about the scientific underpinning of you approach. Clarifications, of the issues that I have raised above, would also be welcome. I don’t think that you need all those cross-sections of vegetation patterns. I suggest that you present a, more richly commented, comparison of one reference section and some characteristic section from the rehabilitated area.

Author Response

Reply to Reviewer 3

Sometimes, I have some difficulties with your language. There are things about which I would need some clarification. Are the whole river stretches rehabilitated, or is it just the urban portions that have been worked on?

☞ Only urban areas have been restored.

You state that you observed transversals along the river(s) between the weirs. What weirs?

☞ In order to reduce misunderstanding, I revised the weir to the embankment. There

are bridges and stone bridges in transverse artificial facilities. In addition,

walkways, bikeways, bare ground, parking lot, etc. were introduced excessively.

What were the time spans between ends of projects and your evaluation?

☞ Rivers that more than five years after restoration lapsed were studied.

There were also two references that made me curious – No 70 and 73 respectively. One is about aquatic ecology, and the other about plant sociology. These are things that you barely, or not at all, deal with in your paper. The list of references also shows many dated items.

☞ 70 is a guideline for river restoration issued by the Korean government, and 73 is

literature containing the methods, which is commonly used for vegetation survey.

River restoration was not such a hot topic in the previous century, as it has become today.

☞ Of course, ecological restoration, including river restoration, is a sort of applied

ecology, which was born in relatively recent days. However, ecology is located in

its background, and restoration, in particular, is based on succession, the process

by which disturbed nature heals itself. In this regard, the investigation method or

basic theory is not much different from the existing ecological theory.

Why did the government undertake these projects?

☞ It was led by the government as the first project to be attempted in Korea.

I doubt that it intended to bring the rivers back to some God given pristine state. However, your evaluation is based on the differences between such a state, and the actual vegetation that you observe and measure. Such a state is, however, not attainable unless you eliminate a great majority of the population, and dynamite the cities. I think that this is, what’s is known as eco-fascism. I am sure that you had more commendable motives. Thus, I think that you need to explain, in some more detail, what or whom that inspired you to undertake this study.

☞ Please refer to the following literatures.

SERI and PWG (Society for Ecological Restoration International Science and Policy Working Group), 2004. The SER International Primer on Ecological Restoration. www.ser.org and Society for Ecological Restoration International, Tucson.

McDonald, T., Gann, G.D., Jonson, J., Dixon, K.W., 2016. International Standards for the Practice of Ecological Restoration-Including Principles and Key Concepts. Washington, DC, Society for Ecological Restoration.

Gann, G.D., McDonald, T., Walder, B., Aronson, J., Nelson, C.R., Jonson, J., Hallett, J.G., Eisenberg, C., Guariguata, M.R., Liu, J., Hua, F., Echeverria, C., Gonzales, E., Shaw, N., Decleer, K., Dixon, K.W., 2019. International principles and standards for the practice of ecological restoration. Restoration Ecology 27, S1-S46.

Speaking of plant sociology, I find it a bit problematic that you do not consider this in your study. There are things that promote, or counteract development of healthy ecosystems. The vegetation patterns that you, more or less prescribe, have developed over a long time, and in locations, whose characteristics may be very different from the ones in the study areas. Young ecosystems are much less resilient. That’s one reason that you found so many garden flowers there. This indicates another omission, that I find problematic. You are so obsessed by the look of the ecosystems, that you forget about their functioning. You mention their flood attenuation capacity, but I think that it’s essentially the floodplain that provides this service.

☞ This study does not address plant sociology. I just applied the survey method used

in plant sociology.

Since rivers are very dynamic spaces, it can be seen that riparian vegetation is

mainly composed of early successional species. In other words, rivers are very

resilient spaces, and even if natural vegetation is introduced for restoration, there

is no big problem in settling. However, introducing plants such as heterogeneous

garden flowers can interfere with the settlement of natural vegetation and increase

flood damage. And the function is maximized when the structure is correctly

configured, and if it is not done correctly, it can cause dysfunction, as shown in the

example of exotic species.

Thus, I suggest that you get some help to make the language a bit more clear. There is a need for a more comprehensive account on why you undertook this study, and about the scientific underpinning of you approach. Clarifications, of the issues that I have raised above, would also be welcome. I don’t think that you need all those cross-sections of vegetation patterns. I suggest that you present a, more richly commented, comparison of one reference section and some characteristic section from the rehabilitated area.

☞ The aim of this study is assessing the ecological effects of river restoration. The

goal of river restoration was improving the overall system of the urban rivers in major

cities in South Korea, which have been under various human interventions for a long time and returning them to a diverse and sustainable ecosystem. Furthermore, another goal of this study is to generate an improvement plan to guide the restored streams toward filling the conditions for successful ecological restoration.

I don't want to restore all these cross-sections of vegetation patterns either. In

particular, I think the space dominated by herbaceous plants with high frequency of

disturbance and strong intensity should be left to the process of nature. This content

was mentioned in the discussion.

Round 2

Reviewer 2 Report

Revision is appropriate